# Transcriptional Regulation of *CD40* Expression by 4 Ribosomal Proteins via a Functional SNP on a Disease-Associated *CD40* Locus

**DOI:** 10.3390/genes11121526

**Published:** 2020-12-21

**Authors:** Meijuan Zou, Xiaoyu Zhang, Danli Jiang, Yihan Zhao, Ting Wu, Qiaoke Gong, Hang Su, Di Wu, Larry Moreland, Gang Li

**Affiliations:** 1Aging Institute at University of Pittsburgh Medical Center, Pittsburgh, PA 15219, USA; MEZ79@pitt.edu (M.Z.); tomlike8161@gmail.com (X.Z.); jiangd@pitt.edu (D.J.); zhaoyihan_7@163.com (Y.Z.); tingwu@pitt.edu (T.W.); qig14@pitt.edu (Q.G.); 2Department of Pharmacology, Nanjing Medical University, Nanjing 211166, China; 3School of Life Science, East China Normal University, Shanghai 200062, China; 4Department of Medicine, Xiangya School of Medicine, Central South University, Changsha 410083, China; 5Bioinformatics and Computational Biology, University of North Carolina, Chapel Hill, NC 27514, USA; hangsu@email.unc.edu; 6Department of Periodontology, University of North Carolina at Chapel Hill, Chapel Hill, NC 27599, USA; did@email.unc.edu; 7Division of Rheumatology, Department of Medicine, University of Pittsburgh, Pittsburgh, PA 15219, USA; larry.moreland@cuanschutz.edu; 8Division of Cardiology, Department of Medicine, University of Pittsburgh, Pittsburgh, PA 15219, USA

**Keywords:** CD40, functional SNP, autoimmune diseases, ribosomal proteins, NF-κB signaling, transcriptional regulation network

## Abstract

Previously, using FREP-MS, we identified a protein complex including eight proteins that specifically bind to the functional SNP (fSNP) rs6032664 at a *CD40* locus associated with autoimmune diseases. Among these eight proteins, four are ribosomal proteins RPL26, RPL4, RPL8, and RPS9 that normally make up the ribosomal subunits involved in the cellular process of protein translation. So far, no publication has shown these ribosomal proteins function as transcriptional regulators. In this work, we demonstrate that four ribosomal proteins: RPL26, RPL4, RPL8, and RPS9 are *bona fide CD40* transcriptional regulators via binding to rs6032664. In addition, we show that suppression of *CD40* expression by *RPL26* RNAi knockdown inactivates NF-κB p65 by dephosphorylation via NF-κB signaling pathway in fibroblast-like synoviocytes (FLS), which further reduces the transcription of disease-associated risk genes such as *STAT4, CD86, TRAF1* and *ICAM1* as the direct targets of NF-κB p65. Based on these findings, a disease-associated risk gene transcriptional regulation network (TRN) is generated, in which decreased expression of, at least, RPL26 results in the downregulation of risk genes: *STAT4, CD86, TRAF1* and *ICAM1,* as well as the two proinflammatory cytokines: *IL1β* and *IL6* via CD40-induced NF-κB signaling. We believe that further characterization of this disease-associated TRN in the CD40-induced NF-κB signaling by identifying both the upstream and downstream regulators will potentially enable us to identify the best targets for drug development.

## 1. Introduction

Autoimmune diseases such as multiple sclerosis (MS), systemic lupus erythematosus (SLE) and rheumatoid arthritis (RA) are complex diseases for which there is no effective cure [1]. The etiology of these diseases is unknown; however, it is publicly recognized that both genetic and environmental factors play a big role in the pathogenesis of these diseases. While it is difficult to identify environmental factors, great progress has been made in discovering the disease-associated genetic factors by using genome wide association studies (GWAS). These studies have identified and, in some cases, validated many genetic loci and risk genes, of which *CD40* is one of the most clinically relevant risk genes for autoimmune diseases [2,3,4].

CD40 is a member of the TNF-receptor superfamily mainly expressed on antigen presenting cells such as B cells as well as other cell types such as FLS [5]. Upon ligation of CD40 with its ligand, CD40L, various signaling pathways are activated including NF-κB, p38, PI3K, and the JAK/STAT pathways. These pathways mediate a broad range of immune and inflammatory responses including T cell-dependent immunoglobulin class switching, memory B cell development, and germinal center formation [6,7,8]. The importance of the CD40-induced NF-κB pathway, especially in RA, has been highlighted based on the involvement of many RA-associated genes in this pathway including *TRAF6*, *TRAF1*, *TNFAIP3*, *NFKBIE*, *REL*, as well as *RBPJ* [4,9]. The same pathway is also highlighted for its role in disease pathogenesis in MS and SLE [10,11]. All of these findings suggest that CD40 could be a potential target for drug development for the treatment of autoimmune diseases. Consistently, blocking the CD40:CD40L axis with various reagents has been proved effective in preventing or ameliorating RA as well as MS and SLE [12,13,14]. Particularly, blocking the interaction between CD40 and CD40L by antibodies targeting CD40L was shown to be highly effective in treating autoimmune diseases; however, it also resulted in thrombotic events due to the targeting of CD40L on platelets, a potential life-threatening side effect that greatly restricted the utility of this therapeutic approach [15,16,17,18,19,20,21,22]. Human genetics suggests that inhibiting intracellular CD40-mediated signaling may be an effective alternative without inducing adverse events related to thrombosis [23]. While some small molecules have been developed to disrupt the binding of CD40 to TRAF6, a CD40 adaptor protein, these compounds have not been tested in humans yet [24,25]. 

In order to identify the more effective targets in the CD40-induced NF-κB signaling pathway, we previously investigated the RA-associated *CD40* locus and identified four fSNPs that regulate *CD40* expression in association with multiple proteins [9]. On one of these fSNPs, rs6032664, eight proteins were identified: RSRC2, HMGB3, FAM76B, SNRPC, RPL26, RPL4, RPL8, and RPS9. Of these, RSRC2 was shown to be a legitimate regulator of *CD40* expression [9]. Superficially, however, it was discouraging that few of the other seven proteins identified suggested that they would be transcriptional regulators. Indeed, four of these proteins (RPL26, RPL4, RPL8, and RPS9) are ribosomal proteins that normally make up the ribosomal subunits involved in the cellular process of protein translation. 

While multiple publications demonstrated that certain ribosomal proteins such as RPL26 are involved in the post-translational regulation of *p53* expression [26], no report has ever demonstrated that ribosomal proteins function as transcriptional regulators, especially regulating disease-associated risk gene expression via binding to causative fSNPs. In this report, by applying RNA interference, luciferase reporter assay, ChIP assay, as well as AIDP-Wb (allele imbalanced DNA pulldown-Western blot), a highly efficient technique to detect allele-specific protein:fSNP binding [27], we demonstrate that *RPL26, RPL4, RPL8* and *RPS9*, together with *HMGB3*, *FAM76B*, and *SNRPC*, are in fact *bona fide CD40* transcriptional regulators via binding to the disease associated fSNP rs6032664. We also demonstrated that these four ribosomal proteins regulate *CD40* expression as a part of a disease-associated TRN in the CD40-induced NF-kB signaling pathway. Thus, by revealing disease-associated TRNs in signaling pathways, we introduce a novel approach to perform post-GWAS functional studies, which will enable us to identify the best targets for drug development.

## 2. Materials and Methods

### 2.1. Cells and Culture

The human B cell line BL2 was purchased from DSMZ (Braunschweig, Germany) with Cat#: ACC 625 and cultured in RPMI1640 medium supplemented with 10% fetal bovine serum, 0.1 M HEPES and 2 mM L-glutamine at 37 °C with 5% CO_2_. The primary FLS was cultured in DMEM medium supplemented with 10% fetal bovine serum. 

### 2.2. Isolation of FLS

Synovial tissue samples were collected from RA patients undergoing knee arthroplasty. All diagnosis of RA met the American College of Rheumatology 1987 revised criteria [28]. The use of human materials is approved by University of Pittsburgh (IRB number STUDY18100138). All methods were performed in accordance with the relevant guidelines and regulations. Written informed consent was obtained from all individual before the operative procedure. FLS were isolated from the synovial tissue samples according to a protocol previously described [29]. 

### 2.3. Primers and Antibodies

All primers listed in Table A1 were purchased from IDT except for *CD40* and *GAPDH*. These primers were purchased from Genecopoeia (Rockville, MD, USA) with Cat#: HQP022955 for the human *CD40* gene and HQP006940 for *GAPDH*. Anti-human antibodies were purchased and used as listed in Table A2.

### 2.4. Western Blot

Whole cell proteins were isolated with RIPA buffer (Sigma, Burlington, MA, USA) (Cat#: R0278) supplemented with 1x EDTA-free protease inhibitor cocktail tablets (Roche, South San Francisco, CA, USA)(Cat#. 11836170001). Cytosolic proteins and nuclear proteins were isolated with NE-PER Nuclear and Cytoplasmic Extraction Reagents (Thermo Fisher Scientific, Waltham, MA, USA)(Cat#: 78833) according to manufacturer’s instructions. Western blots were performed as described [9]. For control, α-tubulin was used for both whole cell proteins and cytosolic proteins and Ku86 for nuclear proteins. The data represents three biological replicates.

### 2.5. AIDP-Wb

Nuclear extracts were isolated as described above. A 31 bp biotinylated double stranded DNA containing either one of the two alleles on rs6032664 was generated from biotinylated primers purchased from IDT by annealing for 5 min at 95 °C following with 30 min incubation at room temperature. 2 μg dsDNA was attached to 20 μL Dynabeads™ M-280 Streptavidin (Invitrogen, Carlsbad, CA, USA)(Cat#: 11206D) according to manufacturer’s instruction. DNA-beads were mixed with ~100 μg nuclear extracts at room temperature for 1 h. with rotation in 60 μL binding buffer used in LightShift^TM^ Chemiluminescent kit (Thermo Fisher Scientific, Waltham, MA, USA)(Cat#: 20148). After magnetic separation and wash, DNA-bound proteins were eluted by adding 1× protein sample buffer and incubate at 95 °C for 5 min. Western blots were performed as described [9]. PARP-1 was used for internal loading control. The data represents three biological replicates.

### 2.6. RNA Isolation 

Total RNA was isolated with RNeasy Mini kit (Qiagen, Germantown MD, USA)(Cat#: 74104). cDNA was synthesized with iScript cDNA Synthesis Kit (BIO-RAD, Hercules, CA, USA)(Cat#: 1708891) after 1 μg RNA sample was treated with DNase I (Invitrogen, Carlsbad CA, USA)(Cat#: 18068015). All the procedures were performed following the manufacturer’s protocols. Real time PCR was done with the StepOne real-time PCR system according to the protocol for the power SYBR green PCR master mix (Applied Biosystems, Foster City, CA, USA)(Cat#: 4367659). For control, GAPDH was used for all the PCR reaction.

### 2.7. RNAi Knockdown 

For shRNA stable knockdown in human BL2 cells, shRNA lentiviruses were obtained from a MISSION^®^ shRNA Library purchased from Sigma (Burlington, MA, USA) (Table A3) and the shRNA knockdown was performed according to the manufacturer’s protocol with a non-mammalian shRNA control. For siRNA transient knockdown in human primary FLS, siRNA targeting *RPL26*, *RPL4*, *RPL8* and *RPS9* was purchased from Thermo Fisher Scientific (Waltham, MA, USA)(Cat#: 4427037-s12202, s12150, s12161, and s12296). FLS were cultured into six-well plate and subjected to transfection when they reached 80% confluency and then siRNA was transfected with Lipofectamine RNAiMAX Transfection Reagent (Thermo Fisher Scientific, Waltham, MA, USA)(Cat#: 13778150) according to the manufacturer’s protocol. Cells transfected with non-targeting siRNA were used as controls.

### 2.8. ChIP Assay

ChIP was performed as described previously [30]. Briefly, scrambled shRNA BL2 cells and *RPL26* shRNA knockdown BL2 cells were cross-linked with 1% formaldehyde for 10 min. Sonication was carried out at 30% amplitude with 20s on and 50s off for 5 min. 10 μg anti-ribosomal protein L26 antibody (Novus Biologicals, Littleton, CO, USA)(Cat#: NB100-2130) coupled to Dynabeads™ Protein A/G (Thermo Fisher Scientific, Waltham, MA, USA)(Cat#:10001D and 10003D) was used for immunoprecipitation. DNA was purified with Qiagen PCR purification kit after reversal of the crosslink. Rabbit IgG was used as an isotype control antibody (Cell signaling Technology, Danvers, MA, USA)(Cat#: 2729). The purified DNA was used for real time PCR analysis of the sequences around the rs6032664. The primers that were used are listed in Table A1.

### 2.9. Luciferase Report Assay 

Luciferase reporter assay was performed by pGL3 Luciferase Reporter Vectors (Promega, Fitchburg, WI, USA)(Cat#: E1751). Luciferase activity was measured by the Dual-Glo^®^ Luciferase Reporter Assay System (Promega, Fitchburg, WI, USA)(Cat#: E2920). All the experiments were performed according to the manufacturer’s protocol. Insert target sequences are listed in Table A1. All data represents a combination of six independent biological replicates. 

### 2.10. Flow Cytometry

The CD40 levels in BL2 cells were measured by FACS analysis with FITC mouse anti-human CD40 (BD Bioscienses; San Jose, CA, USA)(Cat#: 556624). In brief, 0.5 × 10^6^ cells were incubated with 5 μL antibody in PBS/0.02%FBS at 4 °C for 30 min. After washing, FACS was performed as previously described [23] and analyzed using Attune NxT Flow Cytometer Software and FlowJo version 6.

### 2.11. Scratch-Wound Assay

FLS was seeded into 24-well tissue culture plate at a density in ~60–70% confluence. After 24 h. incubation, cells were transfected using siRNA targeting *RPL26, RPL4, RPL8 RPS9, CD40* and *ICAM1*. As a control, a scrambled siRNA transfection was performed in parallel. After another 24 h. incubation, cells were activated by CD40L at 64 ng/ml for 15 min. The monolayer cells were then gently and slowly scratched with a pipette tip across the center of the well. After scratching, the detached cells were gently removed by washing and the well was replenished with fresh medium. After another 24 h. incubation, photos were taken. The data represents three biological replicates.

### 2.12. Statistical Analysis

*P* value was calculated using Student’s T test with 2 tails. Error bars represent the median with S.E.M. For Western blot and AIDP-Wb, the data in each case represents three biological replicates and, for real time PCR, the results represent the combination of three biological replicates unless indicated. 

## 3. Results

### 3.1. Transcriptional Regulation of CD40 Expression by a Protein Complex Containing RPL26, RPL4, RPL8, RPS9, in Human B Cells and FLS

Using SNP-seq and FREP-MS, we previously identified eight proteins that specifically bind to a fSNP rs6032664 on a disease-associated *CD40* locus. Moreover, we demonstrated that one of the eight proteins, RSRC2, was a *CD40* transcriptional suppressor [9]. Among the remaining seven proteins, there are four ribosomal proteins: RPL26, RPL4, RPL8 and RPS9, in addition to HMGB3, FAM76B, and SNRPC. As we know, ribosomal proteins normally make up the ribosomal subunits involved in the cellular process of protein translation. To demonstrate a functional role of these four proteins in *CD40* transcriptional regulation, we performed RNAi knockdown on each of these four ribosomal proteins in BL2, a human B cell line, using a shRNA lentivirus. The role of B cells in the pathogenesis of rheumatoid arthritis has been reported [31]. After viral infection, polyclonal populations, instead of clonal cell lines, were obtained via a puromycin selection to reduce position effects. Total RNA and proteins were isolated from the sample as well as the scrambled shRNA control. Real time PCR analyses and Western blots were performed to evaluate the expression of all these ribosomal proteins as well as CD40. As a result, we observed a significant reduction in the expression of all these four ribosomal proteins on the mRNA (Figure 1A) and protein levels (Figure 1B) and, concomitantly, knockdown of each of these four ribosomal proteins resulted in a significant CD40 down-regulation not only on the protein level as detected by Western blots (Figure 1B), but also on the mRNA level as measured by real time PCR (Figure 1D). We also detected CD40 surface expression on the *RPL26, RPL4, RPL8 and RPS9* shRNA knockdown BL2 cells using FACS analysis and we observed a significant decreased expression of *CD40* in these cells (Figure 1E). These data demonstrate that all these four ribosomal proteins are *CD40* transcriptional regulators. We also performed shRNA knockdown on *HMGB3, FAM76B and SNRPC* in BL2 cells; our data suggest that they are all *CD40* transcriptional regulators (Appendix A
Figure A1).

To demonstrate that the down-regulation of *CD40* expression by the four ribosomal proteins occurred specifically on the transcriptional level, not on the translation level as the function of these proteins in ribosomes would predict, we also checked the expression of *CCR6*, another cell membrane protein in each of these four knockdown cells and no significant change was observed by Western blots (Figure 1B), suggesting that *RPL26, RPL4, RPL8* and *RPS9* specifically regulate *CD40* expression by transcription, not by protein translation in general. 

To validate the function of *RPL26, RPL4, RPL8* and *RPS9* as transcriptional regulators modulating *CD40* expression, we performed an additional experiment. We knocked down each of these four ribosomal proteins in human primary FLS by a transient transfection as shown by both real time PCR (Figure 2A) and Western blots (Figure 2B). In RA, human FLS that lines the joints express inflammatory genes and erosive enzymes that contribute to the pathogenesis of RA [32]. For each knockdown, we used a siRNA that targets a different sequence from that used in shRNA knockdown in human B cells. Consistent with the data shown in Figure 1, we observed a statistically significant decrease in *CD40* expression on mRNA levels detected by real time PCR in all these four FLS (Figure 2D). The downregulation of *CD40* expression was also confirmed on protein levels by Western blots (Figure 2B). Again, to further demonstrate that this regulation of *CD40* expression was carried out on transcriptional level, not on translational level, we checked the expression of *CCR6* by Western blots and no obvious change of *CCR6* expression in these FLS was observed (Figure 2B). 

Here, we notice that, previously when we applied RNAi knockdown on *RSRC2* gene in both human BL2 cells and FLS, we observed an induction of *CD40* expression [9], suggesting that *RSRC2* is a *CD40* transcriptional suppressor. *RSRC2* is one of the eight genes in the complex that bind to the fSNP rs6032664. 

Together, all these data suggest that *RPL26, RPL4, RPL8* and *RPS9* can regulate *CD40* expression on the transcriptional level without changing the overall function of translation as evidenced by the unaltered *CCR6* expression.

### 3.2. Demonstration of the Specific Binding of Ribosomal Protein RPL26, RPL4, RPL8 and RPS9 to fSNP rs6032664 

The ribosomal proteins RPL26, RPL4, RPL8 and RPS9 were originally identified by using FREP-MS with the fSNP rs6032664 [9]. To demonstrate that these proteins function via their binding to the fSNP rs6032664, we first applied AIDP-Wb, a method that was recently developed in our lab [27] to detect the allele-imbalanced binding of a known protein to a specific fSNP. In brief, 31 bp 5′-biotinylated dsDNA fragments centered with either the risk allele T or the non-risk allele A of the fSNP rs6032664 were attached to the equivalent amount of streptavidin-coated Dynabeads. After the DNA-beads were incubated with nuclear proteins isolated from human BL2 cells, nuclear proteins that bind to the two alleles of rs6032664 were magnetically pulled down and analyzed by Western blot using antibodies against each of these four ribosomal proteins. To ensure that the same amount of DNA was used for both alleles, we probed the same Western blot simultaneously using an antibody against PARP-1, a ubiquitous and abundant non-specific dsDNA end-binding protein as internal loading controls. Using this method, a differential binding of RPL26, RPL4, RPL8, and RPS9 to the risk allele T versus the non-risk allele A was observed with the risk allele T having significantly more binding than the non-risk allele A (Figure 3A). At the same time, we also included the binding of RSRC2 to rs6032664 as a positive control since the binding of RSRC2 to this fSNP regulating *CD40* expression was previously demonstrated [9]. Second, we performed a luciferase reporter assay using a luciferase reporter construct that contains the T allele from the fSNP sequence of rs6032664. This construct was used previously together with the construct containing the A allele from the fSNP sequence of rs6032664 to demonstrate the allele-imbalanced luciferase activity for rs6032664 [9]. We performed this luciferase reporter assay in cells having each of these four genes knocked down by siRNA. Our data showed that downregulation of each of these four ribosomal proteins resulted in a significant decrease in luciferase activity (Figure 3B, left). As a negative control, we also performed the same luciferase reporter assay with a luciferase reporter construct containing a SNP sequence from an irrelevant SNP rs7895676 and, in this case, we didn’t observe any significant difference between the samples and the scrambled siRNA control (Figure 3B, right). To further demonstrate that these proteins function via their binding to the fSNP rs6032664, as a proof of evidence, we performed a ChIP assay with the scrambled shRNA WT BL2 cells and the *RPL26* shRNA knockdown cells (Figure 1) using an anti-RPL26 antibody. As a result, we noted a significant enrichment of the fSNP rs6032664 DNA pulled down by the RPL26-specific antibody versus an anti-IgG antibody, suggesting the specificity of the antibody against RPL26. By using this antibody, a significant enrichment of the fSNP rs6032664 sequence was observed in the scrambled shRNA WT BL2 cells versus in the *RPL26* shRNA knockdown BL2 cells (Figure 3C). 

Together, all these data demonstrate the specific binding of RPL26, RPL4, RPL8 and RPS9 to fSNP rs6032664.

### 3.3. A Disease-Associated TRN Linked by CD40-Induced NF-κB Signaling Pathway

Previously, we demonstrated that more CD40 on the surface of B cells (as is the case for carriers of the RA risk allele) has increased activation of the classical NF-κB pathway (as measured by phosphorylation of NF-κB p65) in human B cells upon CD40L activation [23]. To check if decreased *CD40* expression induced by *RPL26* knockdown effects activation of NF-κB p65, we first treated FLS with the siRNA that targets *RPL26* followed by CD40L activation. Activation of NF-κB p65 by CD40L was evaluated with an anti-phosphorylated p65 antibody. As a result, we observed an inactivation of NF-κB p65 by dephosphorylation in the *RPL26* knockdown FLS with a decreased *CD40* expression (Figure 4A). NF-κB has long been known to play an important role in autoimmune diseases such as MS, SLE, and RA [33,34]. Based on TRRUST V2, a manually curated database of human and mouse transcriptional regulatory networks (https://www.grnpedia.org/trrust/Network_search_form.php) [35], we identified all the NF-κB p65 targeted genes. Additionally, using GWAS catalog (https://www.ebi.ac.uk/gwas/), we identified the risk genes that are commonly associated with autoimmune diseases including RA, MS, and SLE. We combined these two results and identified *STAT4, CD86, ICAM1, and TRAF1* as the NF-kB p65 targets that are associated with RA, MS and SLE. Real time PCR was performed on these four genes in the *RPL26* knockdown FLS activated with CD40L. Not to our surprise, we observed a significant downregulation of expression on all these four risk genes in the *RPL26* knockdown FLS in comparison with the scrambled WT controls (Figure 4B). We also checked the expression of IL6 and IL1β, the two proinflammatory cytokines as the direct targets of NF-κB p65 [36,37] also identified by TRRUST V2 and we observed the same downregulation of these two genes as shown in Figure 4C. Together, these data demonstrate that there is a disease-associated risk gene TRN linked by CD40-induced NF-κB signaling in at least human FLS as described in Figure 5A. In this TRN, downregulation of, at least, *RPL26* decreases *CD40* expression, which, in turn, results in the inactivation of NF-κB p65. Inactivation of NF-κB p65 further downregulates *STAT4, TRAF1, ICAM1* and *CD86*, as well as *IL6* and *IL1β* as the direct targets of NF-κB p65.

Among the four risk genes regulated by *RPL26* via CD40-induced NF-κB signaling pathway, ICAM1 is a cell adhesion molecule that is involved in cell migration [38]. In RA, cartilage destruction mediated by invasive FLS plays a central role in pathogenesis of RA and increased FLS migration is fundamental to this process [39]. To test if downregulation of *ICAM1* in FLS induced by *RPL26*, as well as *RPL4, RPL8 or RPS9* knockdown alters FLS migration, we performed a scratch-wound assay, a simple, reproducible assay commonly applied to measure basic cell migration [40]. In the scrambled siRNA WT control, 24 h after being scratched with a pipette tip, the gap is completely closed. However, in *RPL26, RPL4, RPL8* and *RPS9* siRNA knockdown FLS, the gap still remained unfilled as shown in Figure 5B. The same scratch-wound assays were also performed in both *CD40* and *ICAM1* siRNA knockdown FLS and the same results were observed as shown in Figure 5C, which further shows that the defect in cell migration in *RPL26, RPL4, RPL8* and *RPS9* siRNA knockdown FLS might be the consequence of reduced expression levels of *CD40* and *ICAM1*. While these data demonstrate the migration of the FLS is inactivated presumably due to the decreased expression of *ICAM1* triggered by the downregulation of these ribosomal proteins via CD40-induced NF-κB signaling, they also uncover the linkage between the disease-associated risk gene TRN to the intermediate phenotype of the diseases, which further provides an evidence that confirm the existence of the disease-associated TRN linked by CD40-induced NF-κB signaling pathway. 

## 4. Discussion

In this work, we demonstrate that four ribosomal proteins: RPL26, RPL4, RPL8, and RPS9, together with HMGB3, FAM76B, and SNRPC, are *CD40* transcriptional regulators. Ribosomal proteins are correctly believed to play an integral role in protein translation because of their requirement as part of the ribosome. To validate the results from our shRNA knockdown assay in human B cells that ribosomal proteins RPL26, RPL4, RPL8 and RPS9 are *CD40* transcriptional regulators, we performed the same RNAi knockdown assay, but, in a different setting. First, primary human FLS was used, instead of human BL2 cell line; second, siRNA was transfected into FLS by a transient transfection, instead of a stable transfection; third, each siRNA targets a sequence that is different from the sequence that was used in shRNA knockdown in BL2 cells. Consistently enough, both shRNA knockdown in human BL2 cells and siRNA knockdown in human primary FLS on these four ribosomal proteins showed the same results that downregulation of these four ribosomal proteins decreases *CD40* expression on both transcriptional and translational levels (Figure 1 and Figure 2). In addition, a luciferase reporter assay in HMC3 was performed to further demonstrate that all these four ribosomal proteins are transcriptional regulators (Figure 3B). The reason that we used HMC3 is because HMC3 is a brain-resident macrophage that expresses *CD40* and it is easy to perform transfection. 

To demonstrate that *RPL26, RPL4, RPL8 and RPS9* regulate *CD40* expression by transcription, not translation, we checked the expression of *CCR6* in parallel with *CD40* in both shRNA knockdown human BL2 cells and siRNA knockdown human primary FLS. In both cases, we didn’t observe any obvious change of *CCR6* expression in these cells (Figure 1B and Figure 2B). In addition, in all the knockdown experiments on these four ribosomal proteins in both BL2 cells and FLS, we didn’t observe any obvious difference on cellular survival by comparing the samples with the controls (Appendix A
Figure A2), which is in part consistent with the observation that, at least, in Saccharomyces cerevisiae, *RPL26* is not essential for ribosome assembly and function [41]. Together, our data demonstrate that *RPL26, RPL4, RPL8 and RPS9* are the transcriptional regulators modulating *CD40* expression at least in human B cell as well as FLS. These results are consistent with the current findings that ribosomal proteins possess ribosome-independent functions in tumorigenesis, immune signaling, and development [26].

In addition, as we stated above, *RSRC2* is one of eight genes whose products comprise the complex that binds to the fSNP rs6032664. As a newly identified gene, the role of *RSRC2* in transcriptional regulation is not well documented. However, in contrast to the remaining seven genes that were shown as *CD40* transcriptional activators, *RSRC2* was demonstrated as a *CD40* transcriptional suppressor [9]. While our data are intriguing, they fit the competitive mechanism of transcription, in which the activator protein and the repressor protein compete to bind to the same regulatory region of DNA for their function [42]. For example, transcriptional factor Ventx1.1 suppresses its own transcription by binding to a *cis*-acting Ventx1.1 response element (VER) in its own promoter in competition with Xcad2, an activator of *Ventx1.1* transcription [43]; therefore, Xcad2 and Ventx1.1 can competitively occupy VER to regulate *Ventx1.1* transcription in an opposing manner. Currently, we do not have data to show which of the eight proteins in the complex bind directly to the fSNP rs6032664 and whether RSRC2 binds to the fSNP rs6032664 in competition with the four ribosomal proteins. Based on the competitive mechanism, we believe that RSRC2 and the four ribosomal proteins can competitively occupy fSNP rs6032664 to regulate *CD40* transcription in an opposing manner.

Cells are the complex product of gene expression programs involving the coordinated transcription of thousands of genes. Understanding how a collection of diverse regulatory proteins modulates gene expression can best be described using TRN [44,45]. GWAS have identified hundreds of non-coding SNPs that are associated with MS, SLE and RA, but how these SNPs alter gene/protein activity in the context of biological networks remains largely unknown. Indeed, although a handful of alleles have been characterized, in most instances, functional data is missing. By using SNP-seq and FREP-MS, we have been able to collect this critical functional data by not only identifying, but also characterizing fSNPs at the disease-associated *CD40* locus [9]. Although limited, by utilizing this data, as a proof of concept, we are able to generate a disease-associated TRN involving CD40-induced NF-κB signaling (Figure 5A). In this model, novel connections are revealed between the disease-associated *CD40* transcriptional regulators such as *RPL26* and other RA risk genes such as *CD86, STAT4, ICAM1 TRAF1*, and between *RPL26* and proinflammatory cytokines such as *IL6* and *IL1β* (Figure 5). These connections linked via CD40-induced NF-κB signaling presumably could be further extended to many downstream disease-associated intermediate phenotypes such as alteration of cell migration as shown in FLS (Figure 5B). The connection could also be further extended to other risk genes upstream of *CD40*. For example, in the eight proteins that bind to rs6032664, *FAM76B* is also a MS risk gene (GWAS catalog, 2019) (Figure 5A) [9]. However, all these connections need to be carefully validated on both molecular and cellular levels before they can be applied for target identification. 

In summary, we believe that post-GWAS functional studies will enable us to collect the missing functional information on risk gene expression regulation and networking. With all the information, we will be able to generate a disease-associated TRN in a cell type specific fashion for different autoimmune diseases such as MS, RA, and SLE, which should greatly assist us in identifying the best targets for precision drug development.

## Figures and Tables

**Figure 1 genes-11-01526-f001:**
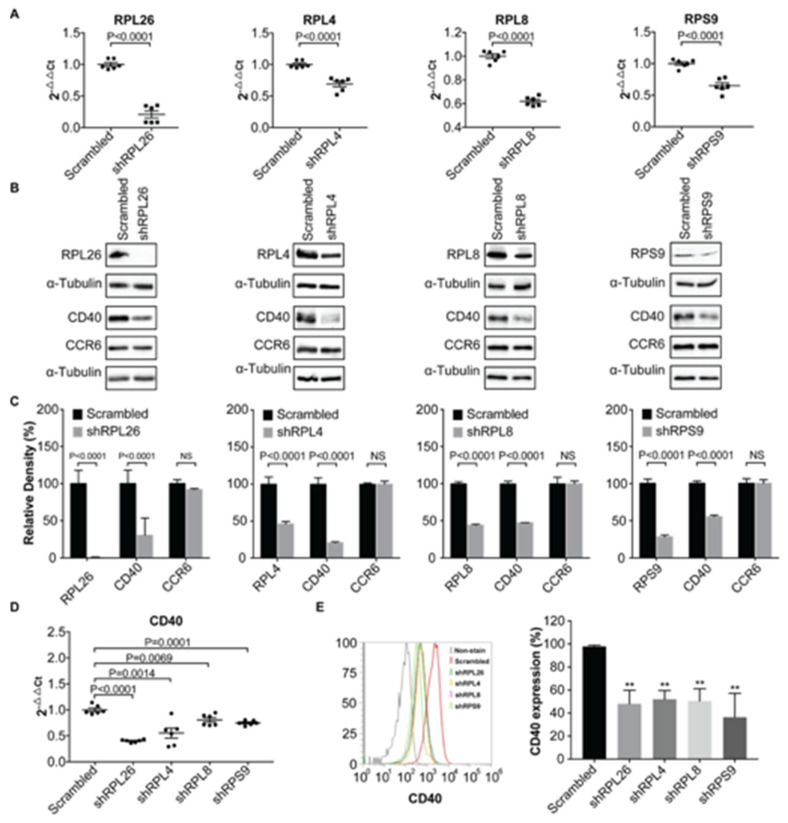
Demonstration of *RPL26, RPL4, RPL8,* and *RPS9*, as *CD40* transcription regulators in human B cells. (**A**) Knockdown of *RPL26, RPL4, RPL8*, and *RPS9*, in human BL2 cells using shRNA as shown by real time PCR. (**B**) Western blots showing downregulation of *RPL26, RPL4, RPL8,* and *RPS9* results in decreased expression of *CD40*, but not *CCR6* in BL2 cells. Whole cell protein was used for Western blots. (**C**) Statistical analysis of Western blots in (**B**) based on relative density. NS: not significant. (**D**) Down-regulation of *CD40* expression on mRNA levels in the knockdown cells, suggesting a transcriptional regulation of *CD40* expression. α-Tubulin was used for loading control. *CCR6* was used as a negative control for translational regulation. (**E**) FACS analysis of CD4*0* surface expression in the *RPL26, RPL4, RPL8,* and *RPS9* shRNA knockdown BL2 cells. Scrambled: scrambled shRNA control. sh: shRNA. **: *p* value < 0.01.

**Figure 2 genes-11-01526-f002:**
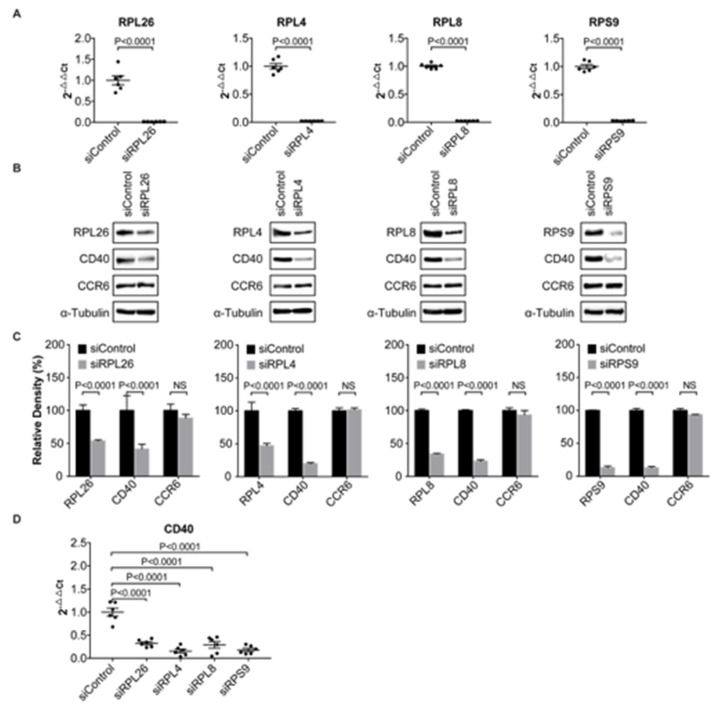
Demonstration of *RPL26, RPL4, RPL8*, and *RPS9* as *CD40* transcription regulators in human FLS. (**A**) Real time PCR showing downregulation of *RPL26*, *RPL4*, *RPL8*, and *RPS9* in FLS by siRNA. (**B**) Western blot showing reduced *CD40* expression in *RPL26, RPL4, RPL8*, and *RPS9* knockdown cells by siRNA. *CCR6* was used as a negative control for translational regulation. (**C**) Statistical analysis of Western blots in (**B**) based on relative density. NS: not significant. (**D**) Real time PCR showing reduced *CD40* expression in downregulation of *RPL26, RPL4, RPL8, and RPS9* in FLS. si: siRNA.

**Figure 3 genes-11-01526-f003:**
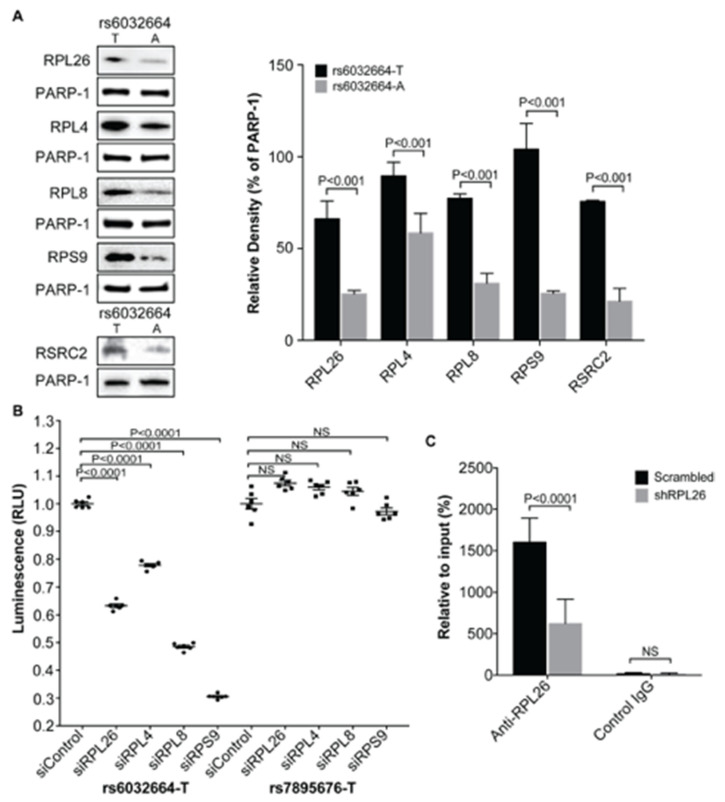
Demonstration of the binding of RPL26, RPL4, RPL8, and RPS9 to rs6032664. (**A**) Allele-imbalanced binding of RPL26, RPL4, RPL8 and RPS9 to the T (risk allele) and A (non-risk allele) of rs6032664 with the statistical analysis based on the relative density. PARP-1 was used as internal loading control and RSRC2 was used as a positive control. (**B**) Luciferase reporter assays with the risk allele T of rs6032664 demonstrate that *RPL26, RPL4, RPL8*, and *RPS9* are the transcriptional regulators. si: siRNA. (**C**). ChIP assay showing endogenous binding of RPL26 to rs6032664. (mean ± SD, n = 3 biological replicates, Student t-test with 2 tails). sh: shRNA. NS: not significant.

**Figure 4 genes-11-01526-f004:**
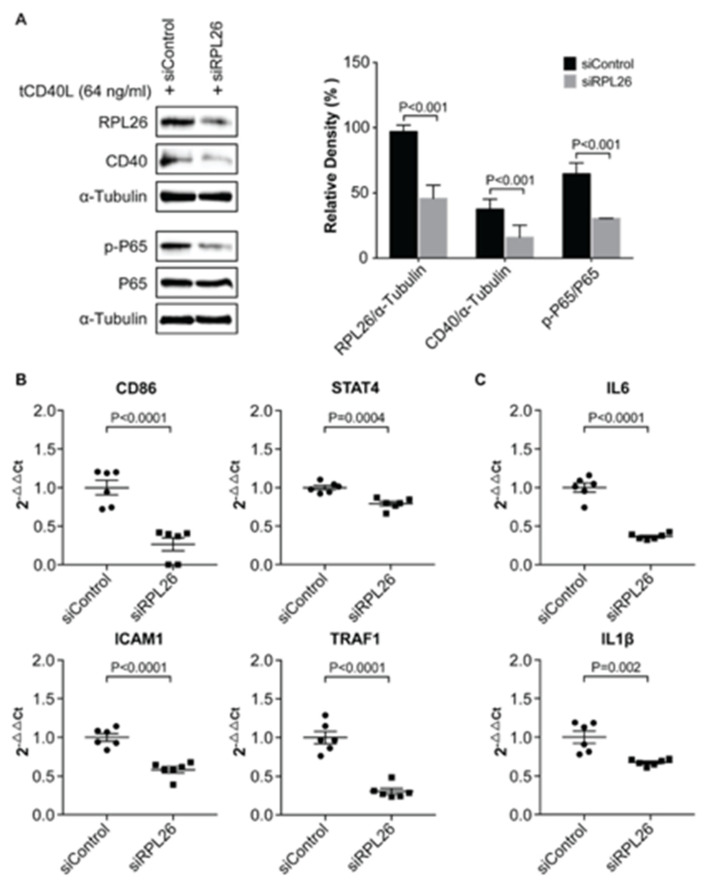
Regulation of disease-associated risk genes *STAT4, TRAF1, ICAM1*, and *CD86* by *RPL26* via disease-associated TRN linked by CD40-induced NF-κB signal pathway. (**A**) Western blots showing inactivation of NF-κB p65 by dephosphorylation upon CD40L activation in the *RPL26* siRNA knockdown FLS. Statistical analysis based on the relative density is shown. (**B**) real time PCR showing downregulation of four disease-associated risk genes *STAT4, TRAF1, ICAM1*, and *CD86* as the direct targets of NF-κB p65, (**C**) real time PCR showing downregulation of IL6 and IL1β as the direct targets of NF-κB p65. si: siRNA.

**Figure 5 genes-11-01526-f005:**
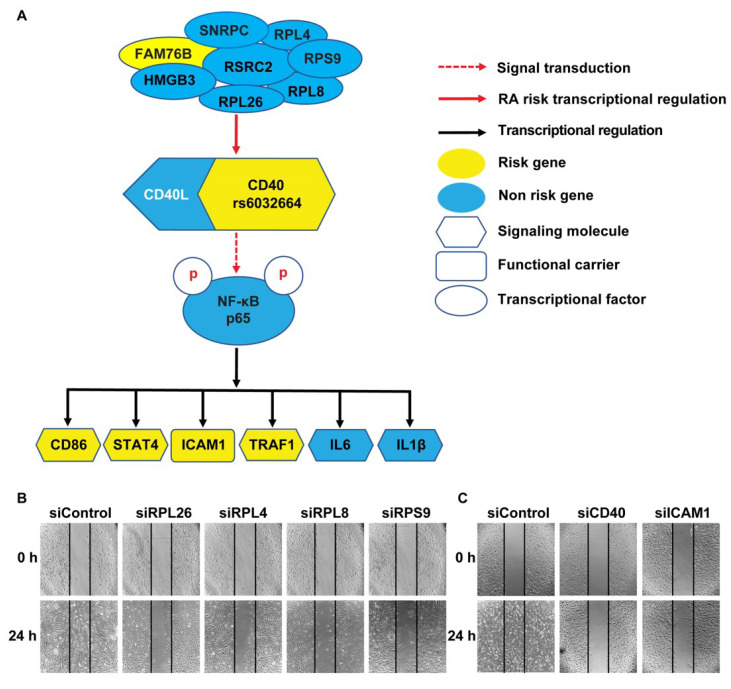
Disease-associated TRN linked by CD40-induced NF-κB signal pathway. (**A**) scheme showing a disease-associated TRN linked by CD40-induced NF-κB signal pathway. Eight proteins including RPL26, RPL4, RPL8, and RPS9 transcriptionally regulate *CD40* expression via binding to fSNP rs6032664. Downstream of CD40-induced NF-κB signaling, *STAT4, TRAF1, ICAM1*, and *CD86* as well as *IL6* and *IL1β* were regulated by NF-κB p65 transcriptionally. *FAM76B* is a risk gene for MS. (**B**) Scratch-wound assay showing defects in cell migration in *RPL26, RPL4, RPL8*, and *RPS9* siRNA knockdown FLS. si: siRNA. (**C**) Scratch-wound assay showing defects in cell migration in FLS with *CD40* and *ICAM1* siRNA knockdown. Downregulation of *CD40* and *ICAM1* was demonstrated by Western blots as shown in Appendix A
Figure A3.

## Data Availability

All data and reagents are available upon reasonable request.

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
