# Peer review of "Transcriptional Regulation of CD40 Expression by 4 Ribosomal Proteins via a Functional SNP on a Disease-Associated CD40 Locus"

_genes, 2020, doi:10.3390/genes11121526_

Round 1

Reviewer 1 Report

Transcriptional regulation of CD40 expression by 4 ribosomal proteins via a functional SNP on a disease-3 associated CD40 locus 

Reviewers comments

In this manuscript, Li and colleagues describe their findings of a protein complex consisting of a total of 8 proteins, including 4 ribosomal proteins, that bind to a functional SNP in the CD40 locus. Binding of this protein complex to the functional SNP leads to an upregulation of the CD40 gene with several downstream effects. The authors show that the mRNA and protein levels of CD40 decreased by downregulating the expression of 7 of these proteins. They clearly presented how these 7 proteins bind to rs6032664 and their higher affinity to the risk allele. Additionally, authors take advantage of transcriptional networks to study the downstream effects of the downregulation of CD40 caused by the KD of the regulative proteins. They show that NFkB signaling and expression of target genes were affected as well. Finally, they showed how one of these downstream genes (ICAM1) can impact a function of the cells such as migration.

The study gives interesting insights in how the CD40 gene is regulated and the consequences of its downregulation. However, there are a number of things that would require additional evidence or a more thorough explanation.

  • Textually, from the manuscript it is not directly clear what the scientific problem is that this study solves and how this is placed in a broader scientific (or disease) context. The study design builds upon previous findings in Li et al., (Nat Genet, 2018) but lacks the introduction for why the regulation of CD40 is interesting in rheumatoid arthritis (RA) and what the implications could be of the findings for this disease. The 8 genes that are found to be in a protein complex that bind to the fSNP are described in Li et al., (Nat Genet, 2018). However, the scientific perspective of this current manuscripts’ rationale is not becoming clear from the text. It is also not clear why more emphasis is put on the ribosomal proteins, rather than the other proteins. These aspects should be better described in the introduction and should also be reflected in the abstract. This will also make a better bridge to the results section. Moreover, some explanations on why certain cell types were used are currently missing.

We find the presented results in this manuscript interesting and the methods that are used are mostly sound. However, some results and claims presented in this manuscript require either additional data or should be explained and discussed better. Moreover, we find some discrepancies with previous work that requires further explanation.

  • Our main concern for the current manuscript is coming from an observation in Li et al., (Nat Genet, 2018). This paper provides the basis for the current manuscript and describes RSRC2, HMGB3, FAM76B, SNRPC, RPL26, RPL4, RPL8, and RPS9 to form a protein complex that bind the fSNP rs6032664 and shows that a KD of RSRC2 (the core component of this complex) causes an UPregulation of the CD40 gene and protein. Figure 1 and 2 of this manuscript, however, show a DOWNregulation of the CD40 gene and protein upon a KD of HMGB3, FAM76B, SNRPC, RPL26, RPL4, RPL8, and RPS9. Because of this conflicting data, it is crucial to see what a KD of RSRC2 would do in these cell systems by including a KD of RSRC2 in figure 1 and 2. One has to have an explanation of why CD40 is oppositely regulated by RSRC2 as compared to HMGB3, FAM76B, SNRPC, RPL26, RPL4, RPL8, and RPS9, which are seemingly all in the same regulatory protein complex, according to Li et al., (Nat Genet, 2018). Also, the RSRC2 protein is missing in figure 5a.

  • In addition to that, in line 216 the authors conclude that the 7 proteins “regulate CD40 expression on the transcriptional level without changing the overall function of translation as evidenced by the unaltered CCR6 expression”. However, concluding overall translation is unaltered based on only unaltered expression levels of CCR6 while ribosomal proteins are knocked down, is too slim evidence for such a conclusion. In the context of all the results of this manuscript and the claim that CD40 expression is regulated on transcriptional level, it is inevitable to prove that translation is not significantly altered. In line 335 it is discussed that “in all the knockdown experiments on these four ribosomal proteins in both BL2 cells and FLS, we didn’t observe any obvious difference on cellular survival by comparing the samples with the controls”. Please include this data, as this would strengthen (but not yet justify) the claim. In order to really prove that translation is unaltered, we propose that the authors should include data of an additional experiment that assesses overall translation while the ribosomal proteins are knocked down (for instance by a simple Puromycin labeling assay as described in Deliu et al., (Biol open., 2017)).

  • Lastly on figure 1 and 2, you see that the KD efficiency of the 7 genes does not seem to correspond to the downregulation of CD40 on protein level. Firstly, it would be good to add quantifications of the western blots to these figures to make quantitative claims. Secondly, do the authors have an explanation for this discrepancy? What are the protein levels of the other proteins in the complex after KD of for example RPL4? Does the complex fall apart completely with only modest KD of RPL4? No other siRNAs used for each target? Off-target siRNA effects possible?

  • In figure 2 and 3, the authors only focus on the ribosomal proteins. What is the reason they disregarded the other proteins from here on? Explanations are missing.

  • In figure 3a, the allele specificity for the T allele was shown for the ribosomal proteins. This is a striking finding. It is however a missed opportunity that this specificity for the T allele is not confirmed in the luciferase reporter and KD assay, as in this experiment there is no plasmid taken along harboring the fSNP with the A allele. Because this information is missing, the evidence for the allele specificity of the 4 ribosomal proteins is too slim to call that “all these data demonstrate the specific binding of RPL26, RPL4, RPL8 and RPS9 to fSNP rs6032664” (line 259).

  • Figure 4 and 5 shows the evidence for a very interesting transcriptional regulatory network, but it needs a better explanation of how the authors exactly identify STAT4, CD86, IL6, ICAM1, TRAF1 and IL1b. Furthermore, the KD consequences on CD40 activation are clear from these figures and the loss of migratory capacity of the FLS is also evident based on the KD of RPL26 and the other ribosomal proteins. However, to fully prove that this is really the consequence of reduced expression levels of CD40 and ICAM1, it would be nice to control this by doing the same experiment with a KD of CD40 and ICAM1 as a positive control (also to further rule out that translation is affected and indirectly causing the phenotype).

  • Finally, in the discussion is written that “a shared interaction of the RA risk genes HSP90AB1, RPS26, RPL6 and RPL3 with RPL26, RPL8 and RPL4 was revealed (unpublished data)”. This is indeed very interesting, but one cannot verify this. It would be good to show some data on this or to leave it out.

References

Li, G. et al. (2018) High-throughput identification of noncoding functional SNPs via type IIS enzyme restriction. Nat 420 Genet. 50(8), 1180-1188.

Deliu, L.P. et al. (2017) Investigation of protein synthesis in Drosophila larvae using puromycin labelling. Biol. Open 6, 1229–1234 

Author Response

The picture can't be displayed here, please find it in the attached file.

Our response to reviewer #1

Textually, from the manuscript it is not directly clear what the scientific problem is that this study solves and how this is placed in a broader scientific (or disease) context. The study design builds upon previous findings in Li et al., (Nat Genet, 2018) but lacks the introduction for why the regulation of CD40 is interesting in rheumatoid arthritis (RA) and what the implications could be of the findings for this disease. The 8 genes that are found to be in a protein complex that bind to the fSNP are described in Li et al., (Nat Genet, 2018). However, the scientific perspective of this current manuscripts’ rationale is not becoming clear from the text. It is also not clear why more emphasis is put on the ribosomal proteins, rather than the other proteins. These aspects should be better described in the introduction and should also be reflected in the abstract. This will also make a better bridge to the results section. Moreover, some explanations on why certain cell types were used are currently missing.

We are sorry that we didn’t make ourselves clear that the major aim of this work is to demonstrate that the four ribosomal proteins RPL26, RPL4, RPL8 and RPS9 are transcriptional regulators that modulate disease-associated CD40 expression via binding to a functional SNP (fSNP) rs6032664 as our title indicates. To our knowledge, this is the first time to demonstrate that ribosomal proteins are transcriptional regulators, which could extend our knowledge on ribosomal proteins beyond their function in protein translation. In addition, by revealing the disease-associated transcriptional regulation network (TRN) linked by CD40 induced NF-κB signaling pathway, we introduce a novel approach to perform post-GWAS functional studies by revealing disease-associated TRN in signal transduction pathways, which provides us a new way to utilize GWAS data for the identification of drug targets for drug development.

We have now revised the abstract in line 23 to 26 and line 34 to 37 and the introduction in line 84 to 87 and 92 to 96 in response to this reviewer’s suggestion. We have also underlined the part for why the regulation of CD40 is interesting in rheumatoid arthritis (RA) and what the implications could be of the findings for this disease. We hope this revision will provide a clear description on the importance and implication of this study for autoimmune diseases such as rheumatoid arthritis.

In addition, we used human B cells and FLS for our studies since both cells express CD40 as described in line 52 now. We also provide the references to demonstrate the important roles these two type cells play in the pathogenesis of RA in line 194 and line 228 to 230.

We find the presented results in this manuscript interesting and the methods that are used are mostly sound. However, some results and claims presented in this manuscript require either additional data or should be explained and discussed better. Moreover, we find some discrepancies with previous work that requires further explanation. Our main concern for the current manuscript is coming from an observation in Li et al., (Nat Genet, 2018). This paper provides the basis for the current manuscript and describes RSRC2, HMGB3, FAM76B, SNRPC, RPL26, RPL4, RPL8, and RPS9 to form a protein complex that bind the fSNP rs6032664 and shows that a KD of RSRC2 (the core component of this complex) causes an UPregulation of the CD40 gene and protein. Figure 1 and 2 of this manuscript, however, show a DOWNregulation of the CD40 gene and protein upon a KD of HMGB3, FAM76B, SNRPC, RPL26, RPL4, RPL8, and RPS9. Because of this conflicting data, it is crucial to see what a KD of RSRC2 would do in these cell systems by including a KD of RSRC2 in figure 1 and 2. One has to have an explanation of why CD40 is oppositely regulated by RSRC2 as compared to HMGB3, FAM76B, SNRPC, RPL26, RPL4, RPL8, and RPS9, which are seemingly all in the same regulatory protein complex, according to Li et al., (Nat Genet, 2018). Also, the RSRC2 protein is missing in figure 5a.

We understand our reviewer’s concern. As the reviewer indicated, previously in Li et al. (Nat Genet, 2018), we demonstrated that RSRC2 is a CD40 transcriptional suppressor. In this manuscript, we demonstrated HMGB3, FAM76B, SNRPC, RPL26, RPL4, RPL8, and RPS9 are the CD40 transcriptional activators even though all these eight proteins bind to the fSNP rs6032664 as a complex. We generated these two pieces of data in the same exact cell systems. We are very confident about these two results.

To consider the reviewer’s suggestion, we have now revised our manuscript in line 238 to 241 to describe this interesting finding with the citation of Li et al. (Nat Genet, 2018).

Here, we will not call these two results as a discrepancy. They are apples and oranges even though they are in the same bag (complex). As we know, many publications have shown that proteins in the same complex function differently, one could be an activator and the other could be a suppressor. For example, in one of our unpublished data, we identified the binding of CUX1, SATB1 and SATB2 as a complex to a fSNP. When we knocked down both CUX1 and SATB1, we found that the risk gene expression is inhibited, however, when we knocked down SATB2, the expression of this risk gene is activated.

We are sorry for the mistake we made in the figure 5a. We have now added RSRC2 in this figure.

In addition to that, in line 216 the authors conclude that the 7 proteins “regulate CD40 expression on the transcriptional level without changing the overall function of translation as evidenced by the unaltered CCR6 expression”. However, concluding overall translation is unaltered based on only unaltered expression levels of CCR6 while ribosomal proteins are knocked down, is too slim evidence for such a conclusion. In the context of all the results of this manuscript and the claim that CD40 expression is regulated on transcriptional level, it is inevitable to prove that translation is not significantly altered. In order to really prove that translation is unaltered, we propose that the authors should include data of an additional experiment that assesses overall translation while the ribosomal proteins are knocked down (for instance by a simple Puromycin labeling assay as described in Deliu et al., (Biol open., 2017)).

We understand our reviewer’s concern on this data. Our rationale on this experiment is based on one-vote veto. If knockdown of RPL26, RPL4 and RPL8 or RPS9 results in a defect in protein translation in general, then we should observe a decreased protein level on every single gene in cells. Since, in all these four knockdown cases, we didn’t observe any obvious change of CCR6, a randomly picked protein, in both B cells and FLS; therefore, we conclude it is not on the translational level. To consider the reviewer’s point, we have now revised the sentence in line 250. Instead of using “demonstrate”, we are using “suggest”.

We also thank our reviewer for the proposal. We agree that Puromycin labeling assay is a method that can detect protein synthesis. However, RPL26, RPL4 and RPL8 and RPS9 are transcriptional regulators. They can alter protein expression via TRN. As we showed in the manuscript, knockdown of each of these four proteins could result in inactivation of transcription factor NF-kB p65 via the CD40-induced NF-kB signaling. In total, there are more than 200 NF-kB p65 target genes, among which many are transcriptional regulators; therefore, theoretically to say, there will be some change on protein levels due to transcriptional regulation network and even though Puromycin labeling assay detects decreased protein levels, it doesn’t mean this is due to a defect of ribosomal proteins in protein translation.

In line 335 it is discussed that “in all the knockdown experiments on these four ribosomal proteins in both BL2 cells and FLS, we didn’t observe any obvious difference on cellular survival by comparing the samples with the controls”. Please include this data, as this would strengthen (but not yet justify) the claim.

We are sorry that we meant that we didn’t observe any obvious difference on cell morphology by comparing the samples with the controls. We have added the data in Appendix figure 2 (line 451) as described in line 381.

Lastly on figure 1 and 2, you see that the KD efficiency of the 7 genes does not seem to correspond to the downregulation of CD40 on protein level. Firstly, it would be good to add quantifications of the western blots to these figures to make quantitative claims. Secondly, do the authors have an explanation for this discrepancy? What are the protein levels of the other proteins in the complex after KD of for example RPL4? Does the complex fall apart completely with only modest KD of RPL4?

First, we thank our reviewer’s advice and we added statistical analysis of the western blots by using densitometry in both Fig. 1C and Fig. 2C.

Second, we also notice the subtle differences of the KD efficiency between the real time PCR and Western blot. But, we don’t think that it is fair to quantitatively compare the results from these two assays since they are apples and oranges. Especially, Western blot is only a semi-quantitative assay. We present the data together from these two assays to show the same KD trend, which demonstrates that all these genes are CD40 transcriptional activators.

We think that our reviewer raised two very good questions here. First, we think that it is unlikely that knockdown of any of these four ribosomal proteins will alter the expression level of other ribosomal proteins unless they are the target genes of that ribosomal protein. In response to the reviewer’s question, we checked the expression of RPL4, RPL8 and RPS9 in RPL26 shRNA knockdown BL2 cells and we didn’t see any obvious change of RPL4, RPL8 and RPS9 (see attached figure below). Second, we don’t think that the complex will fall apart completely with only a modest KD of RPL4. However, of course, more experiments are needed to perfume for clarifying these questions.

No other siRNAs used for each target? Off-target siRNA effects possible?

To rule out off-target effects, we used both shRNA (Fig. 1) and siRNA (Fig. 2) to knock down RPL26, RPL4, RPL8, and RPS9. Since the shRNA target sequence is different from the siRNA target sequence for each gene as described in line 230 to 231 as well as in the discussion in line 368 and 369; therefore, we don’t think that our result is due to the off-target siRNA effects.

In figure 2 and 3, the authors only focus on the ribosomal proteins. What is the reason they disregarded the other proteins from here on? Explanations are missing.

This manuscript, as its title says, aims to study the role of four ribosomal proteins as transcriptional regulators, beyond their function as ribosomal proteins. Therefore, we only performed minimum work on the other three genes HMGB3, FAM76B, and SNRPC. Based on the reviewer’s question, we have now revised our manuscript in section 3.1 in line 206 and 208 and we moved the data we did on these three genes to Appendix figure 1 (line 442). We hope that this will emphasize our work on the four ribosomal proteins.

In figure 3a, the allele specificity for the T allele was shown for the ribosomal proteins. This is a striking finding. It is however a missed opportunity that this specificity for the T allele is not confirmed in the luciferase reporter and KD assay, as in this experiment there is no plasmid taken along harboring the fSNP with the A allele. Because this information is missing, the evidence for the allele specificity of the 4 ribosomal proteins is too slim to call that “all these data demonstrate the specific binding of RPL26, RPL4, RPL8 and RPS9 to fSNP rs6032664” (line 259).

We are sorry that we didn’t mention that the allele-imbalanced luciferase reporter activities of the fSNP rs6032664 were demonstrated in our previous publication (Li et al. Nat. Genet. 2008). In that paper, we used both the T and A alleles from the fSNP rs6032664 for luciferase reporter assay and we observed an allele-imbalanced luciferase reporter activity between the T and A alleles from the fSNP rs6032664. Now we are adding this in our text as shown in line 271 to 274 with the citation.

Figure 4 and 5 shows the evidence for a very interesting transcriptional regulatory network, but it needs a better explanation of how the authors exactly identify STAT4, CD86, IL6, ICAM1, TRAF1 and IL1b. Furthermore, the KD consequences on CD40 activation are clear from these figures and the loss of migratory capacity of the FLS is also evident based on the KD of RPL26 and the other ribosomal proteins. However, to fully prove that this is really the consequence of reduced expression levels of CD40 and ICAM1, it would be nice to control this by doing the same experiment with a KD of CD40 and ICAM1 as a positive control (also to further rule out that translation is affected and indirectly causing the phenotype).

We are sorry that we didn’t explain well on how we identified STAT4, CD86, ICAM1, TRAF1, IL6 and IL1b as the direct targets of NF-kB p65 and how we identified STAT4, CD86, ICAM1, TRAF1 as the risk genes that are associated with autoimmune diseases including RA, MS, and SLE. As we described in line 273 we used TRRUST V2, a manually curated database of human and mouse transcriptional regulatory networks to identify the direct targets of NF-kB p65 and we used GWAS catalog (2019) to identify all the risk genes that are commonly associated with the autoimmune diseases including RA, MS and SLE. Then, we combined these two results and identified STAT4, CD86, ICAM1, and TRAF1 as the direct NF-kB p65 targets that are associated with RA, MS and SLE. We have now listed both the website for TRRUST V2 and for GWAS catalog in line 308 and 309 and described how we identify STAT4, CD86, ICAM1, and TRAF1 in line from 306 to 312. Also, as the reviewer suggests, we performed positive controls for the scratch-wound assays with both CD40 and ICAM1 siRNA knockdown human FLS as described in line 341 to 344 and the data is present in Fig. 5C.

Finally, in the discussion is written that “a shared interaction of the RA risk genes HSP90AB1, RPS26, RPL6 and RPL3 with RPL26, RPL8 and RPL4 was revealed (unpublished data)”. This is indeed very interesting, but one cannot verify this. It would be good to show some data on this or to leave it out.

We identified the shared interaction of the RA risk genes HSP90AB1, RPS26, RPL6 and RPL3 with RPL26, RPL8 and RPL4 by using protein-protein interaction. Since, at this time, we cannot validate this interaction by experiments; therefore, we decide to follow the reviewer’s suggestion and deleted this discussion.

Reviewer 2 Report

In the present work, Zou and colleagues nicely assessed a functional SNP in the vicinity of CD40 which has been previously associated with autoimmune diseases. The authors identified 7 binding proteins that act as transcriptional regulators of CD40 expression in human B cells and in fibroblast-like synoviocytes (FLS). Additionally, the authors generated a transcriptional regulation network inferring the effects of inhibiting the expression of RPL26 in STAT4, CD86, TRAF1, ICAM1, IL1β and IL6 via CD40 induced NF-κB signaling. This is a nice work evaluating the functional effects of an associated variant. To improve the manuscript understanding, the results could be better organized.

Major comments:

1.- Authors should further explain in the manuscript the differential protein binding observed in fSNPs. The allusion to ribosomal proteins is confusing along the text.

2.- The results of the paper rely on previous work where different proteins specifically bind to fSNPs. However, the authors confirmed this in the results section 3.2, where they also showed the allelic imbalance of the binding. For clarity, this should be the first section of the results, reinforcing previous findings. Figure 3B showed the luciferase effects of the T-allele of rs6032664 in knocked down genes, please include the effect of the A-allele. Why were the results reported for knocked down genes? Is there any luciferase effect described for this SNP? This information could be added in the supplementary material.

3.- The immune system is highly redundant, therefore inferring the downregulation of immune related genes only based in the RPL26 knockdown is risky. This should be carefully discussed.

4.- How could these functional results be relevant for other autoimmune diseases that share their genetic risk loci and pathways? Is there a potential use of this analysis for drug repositioning?

Minor comments:

1.- Please revise that abbreviations are defined where first mentioned throughout the manuscript, e.g FLS in the abstract section was not defined.

2.- Gene names should be italicized.

Author Response

Our response to reviewer #2

1.- Authors should further explain in the manuscript the differential protein binding observed in fSNPs. The allusion to ribosomal proteins is confusing along the text.

We fully understand the reviewer’s point. In this manuscript, we did use three different types of experiments including AIDP-Wb, luciferase reporter assay and ChIP assay to further demonstrate the differential protein binding to the fSNP rs6032664 on the CD40 promoter region that is associated with multiple autoimmune diseases. However, the binding of these proteins to the fSNP doesn’t mean that they are functionally relevant unless we can demonstrate that these proteins are really transcriptional regulators modulating CD40 expression and that alteration of these genes’ expression can lead to the phenotypic change of these diseases or, at least, to the change of the intermediate phenotypes of these diseases on the cellular level such as cell migration. In particular, since four of these proteins are ribosomal proteins that normallymake up the ribosomalsubunits involved in the cellular process of protein translationand, based on our knowledge, no report, up to date, has ever demonstrated that ribosomal proteins are transcriptional regulators; therefore, we think that it is extremely important for us to prove that these four ribosomal proteins are the CD40 transcriptional regulators and alteration of their expression can affect cellular function downstream of CD40-induced NF-kB signaling. The success of this work will extend our understanding about the ribosomal proteins beyond their function as ribosomal proteins in protein translation.

2.- The results of the paper rely on previous work where different proteins specifically bind to fSNPs. However, the authors confirmed this in the results section 3.2, where they also showed the allelic imbalance of the binding. For clarity, this should be the first section of the results, reinforcing previous findings. Figure 3B showed the luciferase effects of the T-allele of rs6032664 in knocked down genes, please include the effect of the A-allele. Why were the results reported for knocked down genes? Is there any luciferase effect described for this SNP? This information could be added in the supplementary material.

As we state above, we think that it is equally important to demonstrate both the function of these four ribosomal proteins as transcriptional regulators of CD40 and the allele-imbalanced binding of these protein to the fSNP rs6032664 on the CD40 promoter region that is associated with multiple autoimmune diseases in this manuscript. As a matter of fact, if we cannot demonstrate that these four ribosomal proteins regulate CD40 expression, it will be less interesting to prove the allele-imbalanced binding of these 4 proteins to the fSNP rs6032664 in the CD40 promoter region. Also, our previous work has already demonstrated using CRISPR/cas9 gene editing that this fSNP affects the expression of CD40 in human B cells (Li et al. Nat. Genet. 2020). Base on this rationale, we decide to present the CD40 expression data first in section 3.1 and the binding data second in section 3.2. We hope that our reviewer can reconsider the way we present our data in this manuscript. 

For figure 3B,we are sorry that we didn’t mention that the allele-imbalanced luciferase reporter activities of the fSNP rs6032664 were demonstrated in our previous publication (Li et al. Nat. Genet. 2008). In that paper, we used both the T and A alleles from the fSNP rs6032664 for luciferase reporter assay and we observed an allele-imbalanced luciferase reporter activity between the T and A alleles from the fSNP rs6032664. Now we are adding this in our text as shown in line 271 to 274 with the citation.

  • - The immune system is highly redundant, therefore inferring the downregulation of immune related genes only based in the RPL26 knockdown is risky. This should be carefully discussed.

We completely agree with our reviewer. We revised our manuscript as shown in line 405 and 406 according to the reviewer’s suggestion.

4.- How could these functional results be relevant for other autoimmune diseases that share their genetic risk loci and pathways? Is there a potential use of this analysis for drug repositioning?

This is a very good question. Based on our current knowledge, we think that there are disease-associated CD40-induced NF-kB signal transduction and transcriptional regulationnetworks (STTRN) in a cell type-specific fashion. Even though different autoimmune diseases share the same CD40 signaling, different risk genes in the upstream of CD40 signaling may specifically receive and amplify different environmental signals that activate CD40-induced NF-kB signaling and, at the same time, activation of different risk genes in the downstream of CD40 signaling may result in different effects. Therefore, it depends the set of risk genes each individual has (the genotype), the same environmental factors could trigger different risk genes in different individuals, which could lead to develop different diseases.Obviously, the pathogenesis and/or susceptibility of an autoimmune disease is not due to the signal transduction pathway itself, but the perturbation of a set of risk gene expression, presumably by transcriptional regulation via the risk alleles of many causative fSNPs. When we fully understand the disease-associated STTRN, we will be able to identify the best possible targets for drug development.

Minor comments:

1.- Please revise that abbreviations are defined where first mentioned throughout the manuscript, e.g FLS in the abstract section was not defined.

We have revised all the abbreviations.

2.- Gene names should be italicized.

We have changed this.

Round 2

Reviewer 1 Report

While some additional controls and analysis are done and more references are made to the original article in which the RSRC2 complex that binds rs6032664 is identified, some additional changes should be made to the article.

  • It is clearly stated that RSRC2 is a suppressor in the results, whereas all RP proteins are activators. However, this is a very contradicting result, which should be clearly discussed in the discussion as well. The authors do not conclusively show which protein of the complex binds directly to the SNP and whether the function of the RP proteins is dependent on RSRC2 or vice versa. Also, RSRC2, like the RP proteins seem to be far from a traditional transcription factor but more similar to a splice factor, further complicating the interpretation of the results. While I don’t believe the authors have to tease apart the role of the individual components of the complex and the order of binding to DNA here (although it would be very nice), they should at least address this in the discussion and propose how it is possible that proteins in the same complex have completely opposing effects on CD40 expression.
  • The extra controls in the scratch wound assay are a nice addition, but unfortunately, I cannot clearly see the cells in the pictures, the pictures seem an even grey image to me (could be due to pdf quality).  

Author Response

It is clearly stated that RSRC2 is a suppressor in the results, whereas all RP proteins are activators. However, this is a very contradicting result, which should be clearly discussed in the discussion as well. The authors do not conclusively show which protein of the complex binds directly to the SNP and whether the function of the RP proteins is dependent on RSRC2 or vice versa. Also, RSRC2, like the RP proteins seem to be far from a traditional transcription factor but more similar to a splice factor, further complicating the interpretation of the results. While I don’t believe the authors have to tease apart the role of the individual components of the complex and the order of binding to DNA here (although it would be very nice), they should at least address this in the discussion and propose how it is possible that proteins in the same complex have completely opposing effects on CD40 expression.

We thank our reviewer’s consideration and apologize that we didn’t discuss this in our first revision. We have now added a paragraph to discuss a possible mechanism, the competitive mechanism that could explain why proteins in the same complex could have completely opposing effects on gene expression with an example that was recently published by Kumar et al. (2019) (from line 544 to 558). We believe that RSRC2 and the four ribosomal proteins can competitively bind fSNP rs6032664 to regulate CD40 transcription in an opposing manner.

The extra controls in the scratch wound assay are a nice addition, but unfortunately, I cannot clearly see the cells in the pictures, the pictures seem an even grey image to me (could be due to pdf quality).

We are sorry for the bad quality of Fig. 5C. We have now changed it.